# Precision-Recall-Gain Curves: PR Analysis Done Right

**Peter A. Flach**
Intelligent Systems Laboratory
University of Bristol, United Kingdom
`Peter.Flach@bristol.ac.uk`

**Meelis Kull**
Intelligent Systems Laboratory
University of Bristol, United Kingdom
`Meelis.Kull@bristol.ac.uk`

## Abstract

Precision-Recall analysis abounds in applications of binary classification where true negatives do not add value and hence should not affect assessment of the classifier's performance. Perhaps inspired by the many advantages of receiver operating characteristic (ROC) curves and the area under such curves for accuracy-based performance assessment, many researchers have taken to report Precision-Recall (PR) curves and associated areas as performance metric. We demonstrate in this paper that this practice is fraught with difficulties, mainly because of incoherent scale assumptions – e.g., the area under a PR curve takes the arithmetic mean of precision values whereas the $F_\beta$ score applies the harmonic mean. We show how to fix this by plotting PR curves in a different coordinate system, and demonstrate that the new Precision-Recall-Gain curves inherit all key advantages of ROC curves. In particular, the area under Precision-Recall-Gain curves conveys an expected $F_1$ score on a harmonic scale, and the convex hull of a Precision-Recall-Gain curve allows us to calibrate the classifier's scores so as to determine, for each operating point on the convex hull, the interval of $\beta$ values for which the point optimises $F_\beta$. We demonstrate experimentally that the area under traditional PR curves can easily favour models with lower expected $F_1$ score than others, and so the use of Precision-Recall-Gain curves will result in better model selection.

## 1 Introduction and Motivation

In machine learning and related areas we often need to optimise multiple performance measures, such as per-class classification accuracies, precision and recall in information retrieval, etc. We then have the option to fix a particular way to trade off these performance measures: e.g., we can use overall classification accuracy which gives equal weight to correctly classified instances regardless of their class; or we can use the $F_1$ score which takes the harmonic mean of precision and recall. However, multi-objective optimisation suggests that to delay fixing a trade-off for as long as possible has practical benefits, such as the ability to adapt a model or set of models to changing operating contexts. The latter is essentially what receiver operating characteristic (ROC) curves do for binary classification. In an ROC plot we plot true positive rate (the proportion of correctly classified positives, also denoted *tpr*) on the *y*-axis against false positive rate (the proportion of incorrectly classified negatives, also denoted *fpr*) on the *x*-axis. A categorical classifier evaluated on a test set gives rise to a single ROC point, while a classifier which outputs scores (henceforth called a model) can generate a set of points (commonly referred to as the ROC curve) by varying the decision threshold (Figure 1 (left)).

ROC curves are widely used in machine learning and their main properties are well understood [3]. These properties can be summarised as follows.

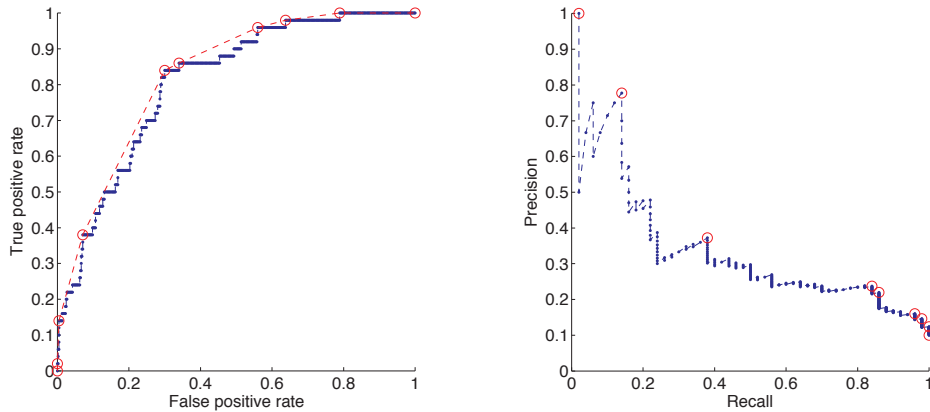

Figure 1: **(left)** ROC curve with non-dominated points (red circles) and convex hull (red dotted line). **(right)** Corresponding Precision-Recall curve with non-dominated points (red circles).

**Universal baselines:** the major diagonal of an ROC plot depicts the line of random performance which can be achieved without training. More specifically, a random classifier assigning the positive class with probability $p$ and the negative class with probability $1 - p$ has expected true positive rate of $p$ and true negative rate of $1 - p$, represented by the ROC point $(p, p)$. The upper-left (lower-right) triangle of ROC plots hence denotes better (worse) than random performance. Related baselines include the always-negative and always-positive classifier which occupy fixed points in ROC plots (the origin and the upper right-hand corner, respectively). These baselines are universal as they don't depend on the class distribution.

**Linear interpolation:** any point on a straight line between two points representing the performance of two classifiers (or thresholds) A and B can be achieved by making a suitably biased random choice between A and B [14]. Effectively this creates an interpolated contingency table which is a linear combination of the contingency tables of A and B, and since all three tables involve the same numbers of positives and negatives it follows that the interpolated accuracy as well as true and false positive rates are also linear combinations of the corresponding quantities pertaining to A and B. The slope of the connecting line determines the trade-off between the classes under which any linear combination of A and B would yield equivalent performance. In particular, test set accuracy assuming uniform misclassification costs is represented by accuracy isometrics with slope $(1 - \pi)/\pi$, where $\pi$ is the proportion of positives [5].

**Optimality:** a point D dominates another point E if D's *tpr* and *fpr* are not worse than E's and at least one of them is strictly better. The set of non-dominated points – the Pareto front – establishes the set of classifiers or thresholds that are optimal under some trade-off between the classes. Due to linearity any interpolation between non-dominated points is both achievable and non-dominated, giving rise to the *convex hull* (ROCCH) which can be easily constructed both algorithmically and by visual inspection.

**Area:** the proportion of the unit square which falls under an ROC curve (*AUROC*) has a well-known meaning as a ranking performance measure: it estimates the probability that a randomly chosen positive is ranked higher by the model than a randomly chosen negative [7]. More importantly in a classification context, there is a linear relationship between $AUROC = \int_0^1 tpr \, dfpr$ and the expected accuracy $acc = \pi tpr + (1 - \pi)(1 - fpr)$ averaged over all possible predicted positive rates $rate = \pi tpr + (1 - \pi)fpr$ which can be established by a change of variable: $\mathbb{E}[acc] = \int_0^1 acc \, drate = \pi(1 - \pi)(2AUROC - 1) + 1/2$ [8].

**Calibration:** slopes of convex hull segments can be interpreted as empirical likelihood ratios associated with a particular interval of raw classifier scores. This gives rise to a non-parametric calibration procedure which is also called isotonic regression [19] or pool adjacent violators [4] and results in a calibration map which maps each segment of ROCCH with slope $r$ to a calibrated score $c = \pi r/(\pi r + (1 - \pi))$ [6]. Define a skew-sensitive version of accuracy as $acc_c \triangleq 2c\pi tpr + 2(1 - c)(1 - \pi)(1 - fpr)$ (i.e., standard accuracy is $acc_{c=1/2}$) then a per-

fectly calibrated classifier outputs, for every instance, the value of $c$ for which the instance is on the $acc_c$ decision boundary.

Alternative solutions for each of these exist. For example, parametric alternatives to ROCCH calibration exist based on the logistic function, e.g. Platt scaling [13]; as do alternative ways to aggregate classification performance across different operating points, e.g. the Brier score [8]. However, the power of ROC analysis derives from the combination of the above desirable properties, which helps to explain its popularity across the machine learning discipline.

This paper presents fundamental improvements in Precision-Recall analysis, inspired by ROC analysis, as follows. (*i*) We identify in Section 2 the problems with current practice in Precision-Recall curves by demonstrating that they fail to satisfy each of the above properties in some respect. (*ii*) We propose a principled way to remedy **all** these problems by means of a change of coordinates in Section 3. (*iii*) In particular, our improved Precision-Recall-Gain curves enclose an area that is directly related to expected $F_1$ score – on a harmonic scale – in a similar way as *AUROC* is related to expected accuracy. (*iv*) Furthermore, with Precision-Recall-Gain curves it is possible to calibrate a model for $F_\beta$ in the sense that the predicted score for any instance determines the value of $\beta$ for which the instance is on the $F_\beta$ decision boundary. (*v*) We give experimental evidence in Section 4 that this matters by demonstrating that the area under traditional Precision-Recall curves can easily favour models with lower expected $F_1$ score than others.

Proofs of the formal results are found in the Supplementary Material; see also `http://www.cs.bris.ac.uk/~flach/PRGcurves/`.

## 2  Traditional Precision-Recall Analysis

Over-abundance of negative examples is a common phenomenon in many subfields of machine learning and data mining, including information retrieval, recommender systems and social network analysis. Indeed, most web pages are irrelevant for most queries, and most links are absent from most networks. Classification accuracy is not a sensible evaluation measure in such situations, as it over-values the always-negative classifier. Neither does adjusting the class imbalance through cost-sensitive versions of accuracy help, as this will not just downplay the benefit of true negatives but also the cost of false positives. A good solution in this case is to ignore true negatives altogether and use precision, defined as the proportion of true positives among the positive predictions, as performance metric instead of false positive rate. In this context, the true positive rate is usually renamed to recall. More formally, we define precision as $prec = TP/(TP + FP)$ and recall as $rec = TP/(TP + FN)$, where $TP$, $FP$ and $FN$ denote the number of true positives, false positives and false negatives, respectively.

Perhaps motivated by the appeal of ROC plots, many researchers have begun to produce Precision-Recall or PR plots with precision on the *y*-axis against recall on the *x*-axis. Figure 1 (right) shows the PR curve corresponding to the ROC curve on the left. Clearly there is a one-to-one correspondence between the two plots as both are based on the same contingency tables [2]. In particular, precision associated with an ROC point is proportional to the angle between the line connecting the point with the origin and the *x*-axis. However, this is where the similarity ends as PR plots have none of the aforementioned desirable properties of ROC plots.

**Non-universal baselines:** a random classifier has precision $\pi$ and hence baseline performance is a horizontal line which depends on the class distribution. The always-positive classifier is at the right-most end of this baseline (the always-negative classifier has undefined precision).

**Non-linear interpolation:** the main reason for this is that precision in a linearly interpolated contingency table is only a linear combination of the original precision values if the two classifiers have the same predicted positive rate (which is impossible if the two contingency tables arise from different decision thresholds on the same model). [2] discusses this further and also gives an interpolation formula. More generally, it isn't meaningful to take the arithmetic average of precision values.

**Non-convex Pareto front:** the set of non-dominated operating points continues to be well-defined (see the red circles in Figure 1 (right)) but in the absence of linear interpolation this set isn't convex for PR curves, nor is it straightforward to determine by visual inspection.

**Uninterpretable area:** although many authors report the area under the PR curve (*AUPR*) it doesn't have a meaningful interpretation beyond the geometric one of expected precision when uniformly varying the recall (and even then the use of the arithmetic average cannot be justified). Furthermore, PR plots have unachievable regions at the lower right-hand side, the size of which depends on the class distribution [1].

**No calibration:** although some results exist regarding the relationship between calibrated scores and $F_1$ score (more about this below) these are unrelated to the PR curve. To the best of our knowledge there is no published procedure to output scores that are calibrated for $F_\beta$ – that is, which give the value of $\beta$ for which the instance is on the $F_\beta$ decision boundary.

## 2.1 The $F_\beta$ measure

The standard way to combine precision and recall into a single performance measure is through the $F_1$ score [16]. It is commonly defined as the harmonic mean of precision and recall:

$$F_1 \triangleq \frac{2}{1/prec + 1/rec} = \frac{2prec \cdot rec}{prec + rec} = \frac{TP}{TP + (FP + FN)/2} \tag{1}$$

The last form demonstrates that the harmonic mean is natural here as it corresponds to taking the arithmetic mean of the numbers of false positives and false negatives. Another way to understand the $F_1$ score is as the accuracy in a modified contingency table which copies the true positive count to the true negatives:

|  | *Predicted* $\oplus$ | *Predicted* $\ominus$ |  |
|---|---|---|---|
| *Actual* $\oplus$ | *TP* | *FN* | *Pos* |
| *Actual* $\ominus$ | *FP* | *TP* | *Neg* − (*TN* − *TP*) |
|  | *TP* + *FP* | *Pos* | 2*TP* + *FP* + *FN* |

We can take a weighted harmonic mean which is commonly parametrised as follows:

$$F_\beta \triangleq \frac{1}{\frac{1}{1+\beta^2}/prec + \frac{\beta^2}{1+\beta^2}/rec} = \frac{(1+\beta^2)TP}{(1+\beta^2)TP + FP + \beta^2 FN} \tag{2}$$

There is a range of recent papers studying the $F$-score, several of which in last year's NIPS conference [12, 9, 11]. Relevant results include the following: (*i*) non-decomposability of the $F_\beta$ score, meaning it is not an average over instances (it is a ratio of such averages, called a pseudo-linear function by [12]); (*ii*) estimators exist that are consistent: i.e., they are unbiased in the limit [9, 11]; (*iii*) given a model, operating points that are optimal for $F_\beta$ can be achieved by thresholding the model's scores [18]; (*iv*) a classifier yielding perfectly calibrated posterior probabilities has the property that the optimal threshold for $F_1$ is half the optimal $F_1$ at that point (first proved by [20] and later by [10], while generalised to $F_\beta$ by [9]). The latter results tell us that optimal thresholds for $F_\beta$ are lower than optimal thresholds for accuracy (or equal only in the case of the perfect model). They don't, however, tell us how to find such thresholds other than by tuning (and [12] propose a method inspired by cost-sensitive classification). The analysis in the next section significantly extends these results by demonstrating how we can identify all $F_\beta$-optimal thresholds for any $\beta$ in a single calibration procedure.

## 3 Precision-Recall-Gain Curves

In this section we demonstrate how Precision-Recall analysis can be adapted to inherit all the benefits of ROC analysis. While technically straightforward, the implications of our results are far-reaching. For example, even something as seemingly innocuous as reporting the arithmetic average of $F_1$ values over cross-validation folds is methodologically misguided: we will define the corresponding performance measure that can safely be averaged.

### 3.1 Baseline

A random classifier that predicts positive with probability $p$ has $F_\beta$ score $(1+\beta^2)p\pi/(p+\beta^2\pi)$. This is monotonically increasing in $p \in [0, 1]$ hence reaches its maximum for $p = 1$, the always-

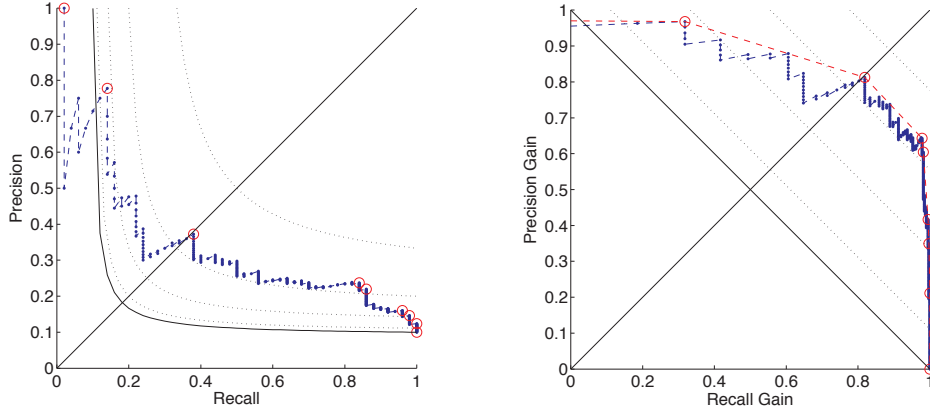

Figure 2: **(left)** Conventional PR curve with hyperbolic $F_1$ isometrics (dotted lines) and the baseline performance by the always-positive classifier (solid hyperbole). **(right)** Precision-Recall-Gain curve with minor diagonal as baseline, parallel $F_1$ isometrics and a convex Pareto front.

positive classifier. Hence Precision-Recall analysis differs from classification accuracy in that *the baseline to beat is the always-positive classifier* rather than any random classifier. This baseline has $prec = \pi$ and $rec = 1$, and it is easily seen that any model with $prec < \pi$ or $rec < \pi$ loses against this baseline. Hence it makes sense to consider only precision and recall values in the interval $[\pi, 1]$. Any real-valued variable $x \in [min, max]$ can be rescaled by the mapping $x \mapsto \frac{x-min}{max-min}$. However, the linear scale is inappropriate here and we should use a harmonic scale instead, hence map to

$$\frac{1/x - 1/min}{1/max - 1/min} = \frac{max \cdot (x - min)}{(max - min) \cdot x} \tag{3}$$

Taking $max = 1$ and $min = \pi$ we arrive at the following definition.

**Definition 1 (Precision Gain and Recall Gain).**

$$precG = \frac{prec - \pi}{(1-\pi)prec} = 1 - \frac{\pi}{1-\pi}\frac{FP}{TP} \qquad\qquad recG = \frac{rec - \pi}{(1-\pi)rec} = 1 - \frac{\pi}{1-\pi}\frac{FN}{TP} \tag{4}$$

A *Precision-Recall-Gain curve* plots Precision Gain on the *y*-axis against Recall Gain on the *x*-axis in the unit square (i.e., negative gains are ignored).

An example PRG curve is given in Figure 2 (right). The always-positive classifier has $recG = 1$ and $precG = 0$ and hence gets plotted in the lower right-hand corner of Precision-Recall-Gain space, regardless of the class distribution. Since we show in the next section that $F_1$ isometrics have slope $-1$ in this space it follows that all classifiers with baseline $F_1$ performance end up on the minor diagonal in Precision-Recall-Gain space. In contrast, the corresponding $F_1$ isometric in PR space is hyperbolic (Figure 2 (left)) and its exact location depends on the class distribution.

## 3.2 Linearity and optimality

One of the main benefits of PRG space is that it allows linear interpolation. This manifests itself in two ways: any point on a straight line between two endpoints is achievable by random choice between the endpoints (Theorem 1) and $F_\beta$ isometrics are straight lines with slope $-\beta^2$ (Theorem 2).

**Theorem 1.** Let $P_1 = (precG_1, recG_1)$ and $P_2 = (precG_2, recG_2)$ be points in the Precision-Recall-Gain space representing the performance of Models 1 and 2 with contingency tables $C_1$ and $C_2$. Then a model with an interpolated contingency table $C_* = \lambda C_1 + (1 - \lambda)C_2$ has precision gain $precG_* = \mu precG_1 + (1 - \mu)precG_2$ and recall gain $recG_* = \mu recG_1 + (1 - \mu)recG_2$, where $\mu = \lambda TP_1 / (\lambda TP_1 + (1 - \lambda)TP_2)$.

**Theorem 2.** $precG + \beta^2 recG = (1 + \beta^2)FG_\beta$, with $FG_\beta = \frac{F_\beta - \pi}{(1-\pi)F_\beta} = 1 - \frac{\pi}{1-\pi}\frac{FP+\beta^2 FN}{(1+\beta^2)TP}$.

$FG_\beta$ is a linearised version of $F_\beta$ in the same way as $precG$ and $recG$ are linearised versions of precision and recall. $FG_\beta$ measures the gain in performance (on a linear scale) relative to a classifier with

both precision and recall – and hence $F_\beta$ – equal to $\pi$. $F_1$ isometrics are indicated in Figure 2 (right). By increasing (decreasing) $\beta^2$ these lines of constant $F_\beta$ become steeper (flatter) and hence we are putting more emphasis on recall (precision).

With regard to optimality, we already knew that every classifier or threshold optimal for $F_\beta$ for some $\beta^2$ is optimal for $acc_c$ for some $c$. The reverse also holds, except for the ROC convex hull points below the baseline (e.g., the always-negative classifier). Due to linearity the PRG Pareto front is convex and easily constructed by visual inspection. We will see in Section 3.4 that these segments of the PRG convex hull can be used to obtain classifier scores specifically calibrated for $F$-scores, thereby pre-empting the need for any more threshold tuning.

## 3.3   Area

Define the *area under the Precision-Recall-Gain curve* as $AUPRG = \int_0^1 precG \, d\,recG$. We will show how this area can be related to an expected $FG_1$ score when averaging over the operating points on the curve in a particular way. To this end we define $\Delta = recG/\pi - precG/(1-\pi)$, which expresses the extent to which recall exceeds precision (reweighting by $\pi$ and $1-\pi$ guarantees that $\Delta$ is monotonically increasing when changing the threshold towards having more positive predictions, as shown in the proof of Theorem 3 in the Supplementary Material). Hence, $-y_0/(1-\pi) \le \Delta \le 1/\pi$, where $y_0$ denotes the precision gain at the operating point where recall gain is zero. The following theorem shows that if the operating points are chosen such that $\Delta$ is uniformly distributed in this range, then the expected $FG_1$ can be calculated from the area under the Precision-Recall-Gain curve (the Supplementary Material proves a more general result for expected $FG_\beta$.) This justifies the use of $AUPRG$ as a performance metric without fixing the classifier's operating point in advance.

**Theorem 3.** Let the operating points of a model with area under the Precision-Recall-Gain curve $AUPRG$ be chosen such that $\Delta$ is uniformly distributed within $[-y_0/1-\pi, 1/\pi]$. Then the expected $FG_1$ score is equal to

$$\mathbb{E}[FG_1] = \frac{AUPRG/2 + 1/4 - \pi(1-y_0^2)/4}{1 - \pi(1-y_0)} \tag{5}$$

The expected reciprocal $F_1$ score can be calculated from the relationship $\mathbb{E}[1/F_1] = (1 - (1-\pi)\mathbb{E}[FG_1])/\pi$ which follows from the definition of $FG_\beta$. In the special case where $y_0 = 1$ the expected $FG_1$ score is $AUPRG/2 + 1/4$.

## 3.4   Calibration

Figure 3 (left) shows an ROC curve with empirically calibrated posterior probabilities obtained by isotonic regression [19] or the ROC convex hull [4]. Segments of the convex hull are labelled with the value of $c$ for which the two endpoints have the same skew-sensitive accuracy $acc_c$. Conversely, if a point connects two segments with $c_1 < c_2$ then that point is optimal for any $c$ such that $c_1 < c < c_2$. The calibrated values $c$ are derived from the ROC slope $r$ by $c = \pi r/(\pi r + (1-\pi))$ [6]. For example, the point on the convex hull two steps up from the origin optimises skew-sensitive accuracy $acc_c$ for $0.29 < c < 0.75$ and hence also standard accuracy ($c = 1/2$). We are now in a position to calculate similarly calibrated scores for $F$-score.

**Theorem 4.** Let two classifiers be such that $prec_1 > prec_2$ and $rec_1 < rec_2$, then these two classifiers have the same $F_\beta$ score if and only if

$$\beta^2 = -\frac{1/prec_1 - 1/prec_2}{1/rec_1 - 1/rec_2} \tag{6}$$

In line with ROC calibration we convert these slopes into a calibrated score between 0 and 1:

$$d = \frac{1}{(\beta^2 + 1)} = \frac{1/rec_1 - 1/rec_2}{(1/rec_1 - 1/rec_2) - (1/prec_1 - 1/prec_2)} \tag{7}$$

It is important to note that there is no model-independent relationship between ROC-calibrated scores and PRG-calibrated scores, so we cannot derive $d$ from $c$. However, we can equip a model with two calibration maps, one for accuracy and the other for $F$-score.

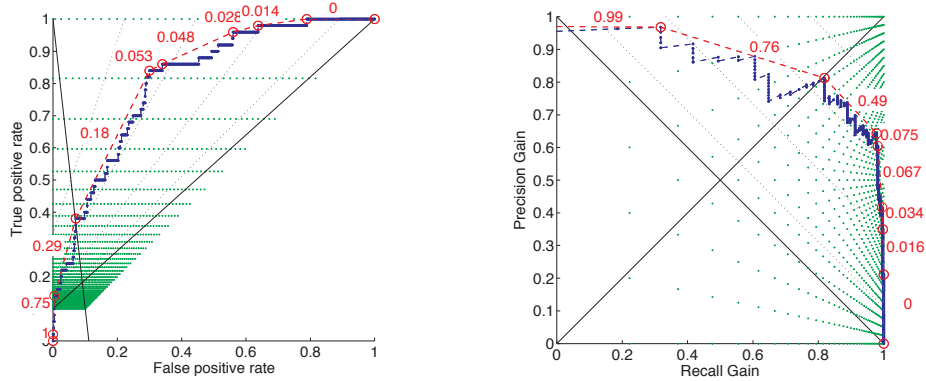

Figure 3: **(left)** ROC curve with scores empirically calibrated for accuracy. The green dots correspond to a regular grid in Precision-Recall-Gain space. **(right)** Precision-Recall-Gain curve with scores calibrated for $F_\beta$. The green dots correspond to a regular grid in ROC space, clearly indicating that ROC analysis over-emphasises the high-recall region.

Figure 3 (right) shows the PRG curve for the running example with scores calibrated for $F_\beta$. Score 0.76 corresponds to $\beta^2 = (1 - 0.76)/0.76 = 0.32$ and score 0.49 corresponds to $\beta^2 = 1.04$, so the point closest to the Precision-Recall breakeven line optimises $F_\beta$ for $0.32 < \beta^2 < 1.04$ and hence also $F_1$ (but note that the next point to the right on the convex hull is nearly as good for $F_1$, on account of the connecting line segment having a calibrated score close to $1/2$).

## 4 Practical examples

The key message of this paper is that precision, recall and $F$-score are expressed on a harmonic scale and hence any kind of arithmetic average of these quantities is methodologically wrong. We now demonstrate that this matters in practice. In particular, we show that in some sense, *AUPR* and *AUPRG* are as different from each other as *AUPR* and *AUROC*. Using the OpenML platform [17] we took all those binary classification tasks which have 10-fold cross-validated predictions using at least 30 models from different learning methods (these are called flows in OpenML). In each of the obtained 886 tasks (covering 426 different datasets) we applied the following procedure. First, we fetched the predicted scores of 30 randomly selected models from different flows and calculated areas under ROC, PRG and PR curves(with hyperbolic interpolation as recommended by [2]), with minority class as positives. We then ranked the 30 models with respect to these measures. Figure 4 plots *AUPRG*-rank against *AUPR*-rank across all 25 980 models.

Figure 4 (left) demonstrates that *AUPR* and *AUPRG* often disagree in ranking the models. In particular, they disagree on the best method in 24% of the tasks and on the top three methods in 58% of the tasks (i.e., they agree on top, second and third method in 42% of the tasks). This amount of disagreement is comparable to the disagreement between *AUPR* and *AUROC* (29% and 65% disagreement for top 1 and top 3, respectively) and between *AUPRG* and *AUROC* (22% and 57%). Therefore, *AUPR*, *AUPRG* and *AUROC* are related quantities, but still all significantly different. The same conclusion is supported by the pairwise correlations between the ranks across all tasks: the correlation between *AUPR*-ranks and *AUPRG*-ranks is 0.95, between *AUPR* and *AUROC* it is 0.95, and between *AUPRG* and *AUROC* it is 0.96.

Figure 4 (right) shows AUPRG vs AUPR in two datasets with relatively low and high rank correlations (0.944 and 0.991, selected as lower and upper quartiles among all tasks). In both datasets *AUPR* and *AUPRG* agree on the best model. However, in the white-clover dataset the second best is AdaBoost according to AUPRG and Logistic Regression according to AUPR. As seen in Figure 5, this disagreement is caused by AUPR taking into account the poor performance of AdaBoost in the early part of the ranking; AUPRG ignores this part as it has negative recall gain.

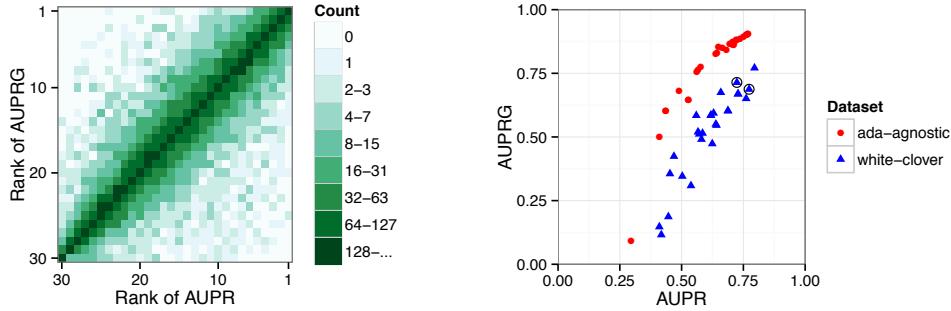

Figure 4: **(left)** Comparison of AUPRG-ranks vs AUPR-ranks. Each cell shows how many models across 886 OpenML tasks have these ranks among the 30 models in the same task. **(right)** Comparison of AUPRG vs AUPR in OpenML tasks with IDs 3872 (white-clover) and 3896 (ada-agnostic), with 30 models in each task. Some models perform worse than random ($AUPRG < 0$) and are not plotted. The models represented by the two encircled triangles are shown in detail in Figure 5.

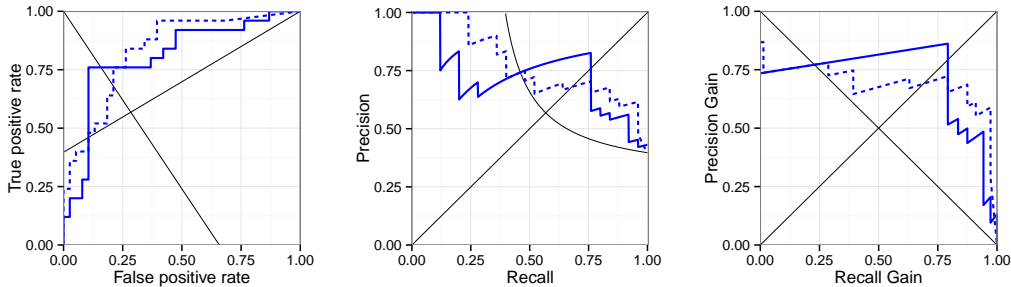

Figure 5: **(left)** ROC curves for AdaBoost (solid line) and Logistic Regression (dashed line) on the white-clover dataset (OpenML run IDs 145651 and 267741, respectively). **(middle)** Corresponding PR curves. The solid curve is on average lower with $AUPR = 0.724$ whereas the dashed curve has $AUPR = 0.773$. **(right)** Corresponding PRG curves, where the situation has reversed: the solid curve has $AUPRG = 0.714$ while the dashed curve has a lower $AUPRG$ of 0.687.

## 5  Concluding remarks

If a practitioner using PR-analysis and the $F$-score should take one methodological recommendation from this paper, it is to use the $F$-Gain score instead to make sure baselines are taken into account properly and averaging is done on the appropriate scale. If required the $FG_\beta$ score can be converted back to an $F_\beta$ score at the end. The second recommendation is to use Precision-Recall-Gain curves instead of PR curves, and the third to use $AUPRG$ which is easier to calculate than $AUPR$ due to linear interpolation, has a proper interpretation as an expected $F$-Gain score and allows performance assessment over a range of operating points. To assist practitioners we have made R, Matlab and Java code to calculate $AUPRG$ and PRG curves available at http://www.cs.bris.ac.uk/~flach/PRGcurves/. We are also working on closer integration of $AUPRG$ as an evaluation metric in OpenML and performance visualisation platforms such as ViperCharts [15].

As future work we mention the interpretation of $AUPRG$ as a measure of ranking performance: we are working on an interpretation which gives non-uniform weights to the positives and as such is related to Discounted Cumulative Gain. A second line of research involves the use of cost curves for the $FG_\beta$ score and associated threshold choice methods.

**Acknowledgments**   This work was supported by the REFRAME project granted by the European Coordinated Research on Long-Term Challenges in Information and Communication Sciences & Technologies ERA-Net (CHIST-ERA), and funded by the Engineering and Physical Sciences Research Council in the UK under grant EP/K018728/1. Discussions with Hendrik Blockeel helped to clarify the intuitions underlying this work.

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
