[Supplementary Material · PRcurves_supplementary.pdf]

# Precision-Recall-Gain Curves:
# PR Analysis Done Right
## *Supplementary Material*

**Peter A. Flach**
Intelligent Systems Laboratory
University of Bristol, United Kingdom
Peter.Flach@bristol.ac.uk

**Meelis Kull**
Intelligent Systems Laboratory
University of Bristol, United Kingdom
Meelis.Kull@bristol.ac.uk

**Theorem 1.** Let $P_1 = (precG_1, recG_1)$ and $P_2 = (precG_2, recG_2)$ be points in the Precision-Recall-Gain space representing the performance of Models 1 and 2 with contingency tables $C_1$ and $C_2$. Then a model with an interpolated contingency table $C_* = \lambda C_1 + (1 - \lambda)C_2$ has precision gain $precG_* = \mu precG_1 + (1 - \mu)precG_2$ and recall gain $recG_* = \mu recG_1 + (1 - \mu)recG_2$, where $\mu = (\lambda TP_1)/(\lambda TP_1 + (1 - \lambda)TP_2)$.

**Proof:** Let us denote $TP_* = \lambda TP_1 + (1 - \lambda)TP_2$ and $FP_* = \lambda FP_1 + (1 - \lambda)FP_2$. Then $\mu = \lambda TP_1/TP_*$ and

$$\mu \frac{FP_1}{TP_1} + (1 - \mu)\frac{FP_2}{TP_2} = \frac{\lambda TP_1}{TP_*}\frac{FP_1}{TP_1} + \frac{(1-\lambda)TP_2}{TP_*}\frac{FP_2}{TP_2} = \frac{\lambda FP_1 + (1-\lambda)FP_2}{TP_*} = \frac{FP_*}{TP_*}.$$

From this it follows that

$$\mu precG_1 + (1-\mu)precG_2 = \mu \left(1 - \frac{\pi}{1-\pi}\frac{FP_1}{TP_1}\right) + (1-\mu)\left(1 - \frac{\pi}{1-\pi}\frac{FP_2}{TP_2}\right)$$

$$= 1 - \frac{\pi}{1-\pi}\left(\mu \frac{FP_1}{TP_1} + (1-\mu)\frac{FP_2}{TP_2}\right) = 1 - \frac{\pi}{1-\pi}\frac{FP_*}{TP_*},$$

but this is equal to $precG_*$ since $FP_*$ and $TP_*$ are entries in the interpolated contingency table $C_*$. The proof for recall gain is identical, with $FN$ instead of $FP$. $\square$

**Theorem 2.** $precG + \beta^2 recG = (1+\beta^2)FG_\beta$, with $FG_\beta = \frac{F_\beta - \pi}{(1-\pi)F_\beta} = 1 - \frac{\pi}{1-\pi}\frac{FP + \beta^2 FN}{(1+\beta^2)TP}$.

**Proof:**

$$precG + \beta^2 recG = 1 - \frac{\pi}{1-\pi}\frac{FP}{TP} + \beta^2\left(1 - \frac{\pi}{1-\pi}\frac{FN}{TP}\right)$$

$$= 1 + \beta^2 - \frac{\pi}{1-\pi}\frac{FP + \beta^2 FN}{TP}$$

$$= (1+\beta^2)\left(1 - \frac{\pi}{1-\pi}\frac{FP + \beta^2 FN}{(1+\beta^2)TP}\right)$$

$$= (1+\beta^2)FG_\beta$$

$\square$

**Theorem 3.** Let $\alpha = 1/(1+\beta^2)$ and $\Delta_\gamma = recG/\pi - precG/\gamma$ with $\gamma \geq 1 - \pi$. Let the operating points of a model with area under the Precision-Recall-Gain curve $AUPRG$ be chosen such that $\Delta_\gamma$ is uniformly distributed within $[-y_0/\gamma, 1/\pi]$. Then the expected $FG_\beta$ score is equal to

$$\mathbb{E}\left[FG_\beta\right] = \frac{(\alpha\gamma + (1-\alpha)\pi)AUPRG + \alpha\pi y_0^2/2 + (1-\alpha)\gamma/2}{\gamma + \pi y_0} \tag{1}$$

**Proof:** First we prove that $\Delta_\gamma$ is monotonically increasing when lowering the threshold $t$ to have more positive predictions. This is needed to calculate expected value of $FG_\beta$ in terms of integrals over $\Delta_\gamma$. For monotonicity we prove that $\Delta_\gamma \leq \Delta'_\gamma$ where $\Delta_\gamma$ and $\Delta'_\gamma$ correspond to thresholds $t$ and $t'$, respectively, with $t > t'$. This holds if and only if:

$$\frac{recG}{\pi} - \frac{precG}{\gamma} \leq \frac{recG'}{\pi} - \frac{precG'}{\gamma} \iff \frac{precG' - precG}{\gamma} \leq \frac{recG' - recG}{\pi}$$

If $recG' = recG$ then this holds, because then $precG' < precG$. Due to $recG' \geq recG$ it is enough to prove that

$$\frac{precG' - precG}{recG' - recG} \leq \frac{\gamma}{\pi}$$

To show this we first note that for any $x > 0$ the equality $\frac{x-\pi}{(1-\pi)x} = \frac{1}{1-\pi}(1 - \pi\frac{1}{x})$ holds, so we have:

$$\frac{precG' - precG}{recG' - recG} = \frac{1/prec - 1/prec'}{1/rec - 1/rec'} = \frac{(FP+TP)/TP - (FP'+TP')/TP'}{\pi n/TP - \pi n/TP'}$$

$$= \frac{FP/TP - FP'/TP'}{\pi n(1/TP - 1/TP')} = \frac{FP(1/TP - 1/TP') - (FP' - FP)/TP'}{\pi n(1/TP - 1/TP')}$$

$$= \frac{FP}{\pi n} - \frac{FP' - FP}{\pi n(TP'/TP - 1)}$$

The first term is upper bounded by $\frac{(1-\pi)n}{\pi n} = \frac{1-\pi}{\pi}$ because the false positives are a subset of all negatives. Since $FP' \geq FP$ and $TP' \geq TP$ due to more positive predictions the subtracted second term cannot be negative. Therefore, we can upper bound this quantity as follows:

$$\frac{precG' - precG}{recG' - recG} \leq \frac{1-\pi}{\pi} + 0 \leq \frac{\gamma}{\pi}$$

where the last inequality is due to $\gamma \geq 1 - \pi$.

This concludes the proof of monotonicity and we can now calculate expected $FG_\beta$ over uniform $\Delta_\gamma$ as follows:

$$\mathbb{E}\left[FG_\beta\right] = \left(\int_{-y_0/\gamma}^{1/\pi} FG_\beta \, d\Delta_\gamma\right) / \left(\int_{-y_0/\gamma}^{1/\pi} d\Delta_\gamma\right)$$

We have $FG_\beta = (1-\alpha)recG + \alpha precG$ and so

$$\mathbb{E}\left[FG_\beta\right] = \left(\int_{-y_0/\gamma}^{1/\pi} ((1-\alpha)\pi recG/\pi + \alpha precG - (1-\alpha)\pi precG/\gamma + (1-\alpha)\pi precG/\gamma)\, d\Delta_\gamma\right) / (1/\pi + y_0/\gamma)$$

$$= \left(\int_{-y_0/\gamma}^{1/\pi} ((1-\alpha)\pi\Delta_\gamma + (\alpha + (1-\alpha)\pi/\gamma)precG)\, d\Delta_\gamma\right) \pi\gamma / (\gamma + \pi y_0)$$

$$= \frac{(1-\alpha)\pi^2\gamma}{\gamma + \pi y_0} \int_{-y_0/\gamma}^{1/\pi} \Delta_\gamma d\Delta_\gamma + \frac{\gamma\pi\alpha + (1-\alpha)\pi^2}{\gamma + \pi y_0} \int_{-y_0/\gamma}^{1/\pi} precG \, d\Delta_\gamma$$

$$= \frac{(1-\alpha)\pi^2\gamma}{\gamma + \pi y_0}(1/\pi^2 - y_0^2/\gamma^2)/2 + \frac{\gamma\pi\alpha + (1-\alpha)\pi^2}{\gamma + \pi y_0} \int_0^1 precG \frac{d\Delta_\gamma}{d\, recG} d\, recG$$

Since $\frac{d\Delta_\gamma}{d\, recG} = \frac{1}{\pi} - \frac{1}{\gamma}\frac{d\, precG}{d\, recG}$, we can rewrite the integral as follows:

$$\int_0^1 precG \frac{d\Delta_\gamma}{d\, recG} d\, recG = \frac{1}{\pi} \int_0^1 precG \, d\, recG - \frac{1}{\gamma} \int_0^1 precG \frac{d\, precG}{d\, recG} d\, recG$$

$$= \frac{1}{\pi} AUPRG - \frac{1}{\gamma} \int_{y_0}^0 precG \, d\, precG = \frac{1}{\pi} AUPRG + \frac{1}{\gamma} y_0^2/2$$

Therefore,

$$
\begin{aligned}
\mathbb{E}\left[FG_\beta\right] &= \frac{(1-\alpha)\pi^2\gamma}{\gamma+\pi y_0} \cdot \frac{\gamma^2-\pi^2 y_0^2}{2\gamma^2\pi^2} + \frac{\gamma\pi\alpha+(1-\alpha)\pi^2}{\gamma+\pi y_0}\left(\frac{1}{\pi}AUPRG + \frac{1}{\gamma}{y_0}^2/2\right) \\
&= \frac{(1-\alpha)(\gamma^2-\pi^2 y_0^2)}{2\gamma(\gamma+\pi y_0)} + \frac{\gamma\pi\alpha y_0^2+(1-\alpha)\pi^2 y_0^2}{2\gamma(\gamma+\pi y_0)} + \frac{\gamma\alpha+(1-\alpha)\pi}{\gamma+\pi y_0}AUPRG \\
&= \frac{(1-\alpha)\gamma^2+\gamma\pi\alpha y_0^2}{2\gamma(\gamma+\pi y_0)} + \frac{\alpha\gamma+(1-\alpha)\pi}{\gamma+\pi y_0}AUPRG \\
&= \frac{\alpha\gamma+(1-\alpha)\pi}{\gamma+\pi y_0}AUPRG + \frac{\alpha\pi y_0^2+(1-\alpha)\gamma}{2(\gamma+\pi y_0)} \\
&= \frac{(\alpha\gamma+(1-\alpha)\pi)\left(AUPRG+\alpha\pi y_0^2/2+(1-\alpha)\gamma/2\right)}{\gamma+\pi y_0}
\end{aligned}
$$

$\square$

**Corollary.** *Under uniform $\Delta_\gamma$ for $\gamma = 1-\pi$ the expected $FG_1$ equals to the following:*

$$
\mathbb{E}\left[FG_1\right] = \frac{AUPRG/2+1/4-\pi(1-y_0^2)/4}{1-\pi(1-y_0)}
$$

**Theorem 4.** Let two classifiers be such that $prec_1 > prec_2$ and $rec_1 < rec_2$, then these two classifiers have the same $F_\beta$ score if and only if

$$
\beta^2 = -\frac{1/prec_1 - 1/prec_2}{1/rec_1 - 1/rec_2} \tag{2}
$$

**Proof:** The slope of the line segment connecting the two classifiers in PRG space is

$$
\frac{precG_1 - precG_2}{recG_1 - recG_2} = \frac{(1/prec_1 - 1/\pi) - (1/prec_2 - 1/\pi)}{(1/rec_1 - 1/\pi) - (1/rec_2 - 1/\pi)}
$$

according to the first expression in Equation 3 in the main paper (the denominators cancel out). This slope is equal to $-\beta^2$ according to Theorem 2 and establishes a line of constant $FG_\beta$ and hence constant $F_\beta$. $\square$