[Reviews · NeurIPS 2015]

Submitted by Assigned_Reviewer_1

(1) Another important disadvantage of recall precision curves is that they are not invariant with respect to the distribution of labels in the test data (proportion \pi). This should be also discussed in the paper. (2) Figure 5 should be included in the main paper, since it directly compares the ranking with respect to PR and PRG. (3) PR curves and average precision are standard tools for evaluating object detection approaches, e.g. for the Pascal VOC Challenge. It would be very interesting if the authors would be able to get access to the results in the challenge and rerank the entries of the competition according to precision recall gain curves. This would also increase the expected impact of the paper.
Summary: This paper is easy to read and offers interesting insights important for many ML people. I really enjoyed reading it.

Submitted by Assigned_Reviewer_2

The paper presents a Precision-Recall-Gain curves to evaluate the precision recall relationship in an efficient way. The analysis of, 1) baselines, 2) interpolation, 3) area under curve, 4) calibration and 5) optimality have been developed and explained for the introduced PRG curve. The paper suffers from lack of well experimental results presentation which have been posted to supplementary documents.

The writing needs more work in order to make it easy to read and follow.

Here is my comments and questions: 1- What is the last column in Contingency table in page 4? The TP at the third column should be TN?

2- What is x in line 243? I think it corresponds to a point in the PR curve. You need to explain any term when in used for the first time.

3- The figures are not clear in BW print.

4- Where is figure 5? You can't address any figure in supplementary documents in the paper.

Summary: Interesting practical paper. However, the paper is not self-explanatory and hard to understand without supplementary documents. Writing need more work too.

Submitted by Assigned_Reviewer_3

Minor points *I think it would be able to try to give a sense of what precision gain and recall gain mean intuitively.

*I find the captions in the figures a bit confusing and difficult to work through. I think that this would be clearer if there were separate or sub figures.
Summary: I think that this is an interesting paper as it increases our understanding of precision-recall space. It addresses important open issues about the PR space and should be published.

Submitted by Assigned_Reviewer_4

These authors demonstrate that traditional Precision-Recall (PR) curves and associated area under PR curve are fraught with difficulties.

They show how to fix this by plotting PR curves in a different coordinate system, and propose Precision-Recall-Gain curves which inherit all key advantages of ROC curves. This study is of interest to the readership and suitable for the NIPS audience. This innovative study will advance the field of performance assessment for predictive models and diagnostic tests. However, several problems are noted in the methods and clarity of presentation.

a. At line 390, what are AUROCs for IBK and Logistic Regression? b. It is difficult to interpret the information embedded in Figure 5.

I would recommend using a table to report the outcomes for five Weka models trained on 19 UCI datasets, and using Figure 5 to display performance for several examples including the mushroom dataset.

c. For Figure 5, it is better to avoid abbreviations and to provide detailed explanation so that it can stand alone. It is also better to provide the definition for IBK since it isn't a widely used model.
Summary: These authors demonstrate that traditional Precision-Recall (PR) curves and associated area under PR curve are fraught with difficulties.

They show how to fix this by plotting PR curves in a different coordinate system, and propose Precision-Recall-Gain curves which inherit all key advantages of ROC curves.

Author Feedback
Author rebuttal: We thank the reviewers for their insightful comments, and will address the issues raised and questions asked as much as possible in this author response.

1. Several reviewers made comments about the figures, and in particularly Fig.5. In the final version we will (i) make sure all figures are clear in B/W; (ii) provide separate captions for the left and right figures; (iii) move Fig.5 to the main paper; (iv) clearly explain all models used in the experiment.

2. Another issue requiring clarification was raised by R4:
"david hand discusses other problems from a paper 6 years ago:
http://link.springer.com/article/10.1007%2Fs10994-009-5119-5#page-1
in other words, ROC curves and AUC is useful, but known to be problematic."

Thanks for raising this. First, note that Hand only criticised AUC as a measure of classification performance (not ROC curves as such, nor AUC as a measure of ranking performance). In his paper Hand set out to relate AUC to expected misclassification loss under varying cost proportions, but the derived relation was model-dependent as it assumed optimal decision thresholds only. It has since been shown that if thresholds are aggregated in a different way the model dependence vanishes and AUC can be translated into expected loss, see citation [7] in the paper. We will add a clarification to the paper. Note that Theorem 2 of our paper uses similar ideas to relate AUPRG to expected F1 score.

3. (R1) "Another important disadvantage of recall precision curves is that their not invariant with respect to the distribution of labels in the test data (proportion \pi). This should be also discussed in the paper."

This was intended to be captured under 'Non-universal baselines' at the top of p.4. This will be emphasised more.

4. (R1) "PR curves and average precision are a standard tool for evaluating object detection approaches, e.g. for the Pascal VOC Challenge. It would be very interesting if the authors would be able to get access to the results in the challenge and rerank the entries of the competition according to precision recall gain curves."

This is an excellent suggestion, we have already contacted the organisers of the VOC Challenge.

5. (R2) "What is the last column in Contingency table in page 4? The TP at the third column should be TN?"

The last column and the last row give row/column marginals, this will be clarified in the paper. TP in the third column is correct, this reflects the fact that F-score ignores TN and instead weights TP twice compared to FP and FN, see Eq.(1).

6. (R2) "What is x in line 243?"

Here x refers to an arbitrary value measured on a scale from min to max, which is then mapped to the [0,1] interval by Eq.(3). We use this mapping to normalise prec, rec, and F_beta. This will be clarified.

7. (R3) "At line 390, what are AUROCs for IBK and Logistic Regression?"

The numbers are as follows (these will be added to the paper):

model = Logistic
AUPR = 0.862069183648017
AUPRG = 0.864815307546806
AUROC = 0.934834227911619

model = IBk
AUPR = 0.911434951724443
AUPRG = 0.837058453292835
AUROC = 0.870355895866427

8. (R6) "I think it would be able to try to give a sense of what precision gain and recall gain mean intuitively."

Eq.(3) on p.5 was meant to convey this intuition (unfortunately it contains a small mistake: the numerator of the right-hand side should read max*(x-min) rather than max*x-min. Since max=1 in all applications of this equation this doesn't affect subsequent results). The key idea is to normalise using a harmonic scale rather than a linear scale, since precision and recall are ratios.